# Assessing Industrial Communication Protocols to Bridge the Gap between Machine Tools and Software Monitoring

**DOI:** 10.3390/s23125694

**Published:** 2023-06-18

**Authors:** Endika Tapia, Leonardo Sastoque-Pinilla, Unai Lopez-Novoa, Iñigo Bediaga, Norberto López de Lacalle

**Affiliations:** 1Aeronautics Advanced Manufacturing Center, University of the Basque Country, 48940 Leioa, Spain; edwarleonardo.sastoque@ehu.eus (L.S.-P.); unai.lopez@ehu.eus (U.L.-N.); norberto.lzlacalle@ehu.eus (N.L.d.L.); 2Department of Computer Languages and Systems, University of the Basque Country, 48940 Leioa, Spain; 3Ideko, 20870 Elgoibar, Spain; ibediaga@ideko.es; 4Department of Mechanical Engineering, University of the Basque Country, 48940 Leioa, Spain

**Keywords:** performance evaluation, monitoring, industrial communication protocols, data acquisition, machine tools

## Abstract

Industrial communication protocols are protocols used to interconnect systems, interfaces, and machines in industrial environments. With the advent of hyper-connected factories, the role of these protocols is gaining relevance, as they enable the real-time acquisition of machine monitoring data, which can fuel real-time data analysis platforms that conduct tasks such as predictive maintenance. However, the effectiveness of these protocols is largely unknown and there is a lack of empirical evaluation which compares their performance. In this work, we evaluate OPC-UA, Modbus, and Ethernet/IP with three machine tools to assess their performance and their complexity of use from a software perspective. Our results show that Modbus provides the best latency figures and communication has different complexities depending on the used protocol, from the software perspective.

## 1. Introduction

The adoption of emerging technologies such as the Industrial Internet of Things (IIoT), Big Data, and Artificial Intelligence (AI) is experiencing notable growth in organizations. This trend aims to enhance productivity and increase competitiveness in the market. These technologies enable the collection of large volumes of data and the application of advanced analytical algorithms to obtain valuable real-time insights. In highly interconnected environments, efficiency and quality heavily rely on data acquisition and the implementation of predictive models, making these technologies essential components [1,2].

Figure 1 represents the different stages in a platform oriented to data analysis. The first stage, which is one of the main focuses of this article, is centered around acquiring relevant data at the required frequencies, which entails determining how to access them. Once the data are collected, they need to be processed and transformed for subsequent analysis and model training. Next, there is the data publication stage, which may involve using the model for making predictions or visualization purposes. Finally, data storage ensures their reuse in the future.

However, despite its tremendous potential, there are still obstacles that hinder its widespread adoption in numerous companies, as indicated in the European Commission report on the European Data Strategy [3]. One of the main reasons for this situation is the complexity associated with extracting production data, especially in advanced manufacturing environments where real-time analytics are sought. The complexity of this challenge stems from the diversity of sources and industrial communication protocols (ICPs) available in each of them, the lack of standardization, the lack of interoperability between systems and equipment, as well as the variability in data formats and quality.

Machine tools in advanced manufacturing environments are usually equipped with a Programmable Logic Controller (PLC), an embedded computer responsible for managing automation logic and data handling, among others. The demand for connectivity of these PLCs has been increasing to meet the current industry requirements [4], through the use of available ICPs. These protocols enable connectivity between machines, equipment, and systems as part of an industrial network, providing comprehensive visibility and control over activities carried out in the production area. To leverage machine data and harness the potential of Industry 4.0 applications, it is essential to acquire data with appropriate latency based on the targets of the application. However, the challenges to efficiently use industrial networks and protocols remain to be properly laid out.

### Related Work

When conducting bibliographic research in the scientific literature, it becomes evident that there is a scarcity of research conducting empirical evaluations of the performance of ICPs such as Open Platform Communications Unified Architecture (OPC-UA) or Ethernet/IP. Among the few studies carried out is the work of Wang et al. [5], who compared the performance between OPC-UA and Messaging Queuing Telemetry Transport (MQTT) in terms of packet overhead, latency, packet loss and CPU usage, demonstrating that MQTT delivers better performance in terms of packet overhead and CPU utilization. In a related study, Rocha et al. [6] performed a performance comparison between OPC-UA and MQTT, focusing on the overall data transfer quantity, including user payload and overhead, as well as the roundtrip time for data exchange and feedback. Their experimental analysis encompassed diverse cloud computing server and application scenarios. The findings of the study also revealed that MQTT exhibited faster performance than OPC UA specifically for pure data exchange.

Furthermore, Cavaleri and Cutuli [7] conducted a study proposing measurement parameters for evaluating the performance of OPC UA. Their research emphasized the importance of key features such as security, subscription mechanisms, and sampling intervals. Imtiaz and Jasperneite [8] investigated OPC-UA as a middleware solution for resource-limited devices. They implemented an OPC-UA server based on the “Nano Embedded Device Server profiles” of the OPC Foundation, demonstrating that OPC-UA can scale down to a chip level while retaining its prominent features and usefulness.

On the other hand, there are numerous studies that compare different characteristics of ICPs but do not conduct empirical performance evaluations. For instance, Anitha et al. [9] analyzed and compared the implementation of different protocols based on network topology (HTTP, MQTT, CoAP, XMPP, AMQP, and Modbus), along with their computational and latency performance. Petkov and Naumov [10] described the industrial communications used in process automation, the challenges and practical topologies of automation networks, and technological perspectives. Additionally, Vituri et al. [11] provided a review of industrial communication networks, addressing the state of the art and outlining interesting future perspectives. Meanwhile, Lin and Pearson [12] examined various industrial ethernet protocols and advocate for the need to create a unified hardware and software platform upon which multiple standards can be implemented, offering real-time communication, determinism, and low latency.

Ensuring the security of communication protocols is of paramount importance to protect industrial systems and sensitive data from unauthorized access and malicious attacks. Gurtov et al. [13] emphasized the challenge of achieving ubiquitous connectivity for sensors in noisy industrial environments, highlighting the need to collect and securely transmit sensor data for further processing in cloud storage or smart spaces.

In the context of the IoT paradigm, attention to edge computing becomes essential to address both security and intelligence requirements. Besednyi et al. [14] proposed an edge computing model that focuses on receiving and processing sensor data, utilizing specialized computing modules to gather raw data from multiple sensors in the physical environment. Their study highlights the advantages of implementing local data processing on edge or intermediate devices, improving security and communication efficiency by leveraging advanced security measures and minimizing the transmission of sensitive data.

This work extends beyond the existing literature, presenting an empirical study of three ICPs: OPC-UA, Modbus, and Ethernet/IP. It aims to answer research questions about how these protocols differ in terms of performance and complexity of use from a software perspective, and how can their effectiveness be empirically evaluated and compared.

The evaluation has been conducted in the Aeronautics Advanced Manufacturing Center [15] (or CFAA, for its initials in Spanish), a research center dedicated to the development of advanced manufacturing technologies for aeronautical engine components and other aerospace elements. The CFAA conducts applied research between Technology Readiness Levels (TRL) 5 and 7 [16] with technologies including high-precision machining [17], broaching [18], or additive manufacturing [19]. The CFAA has state-of-the-art machining centers from various manufacturers, including Fagor, Danobat, and Hermle.

The remainder of this paper is structured as follows: Section 2 describes and compares the protocols studied in this work, and Section 3 presents software tools to enable data acquisition from general purpose computing systems. Section 4 presents the results of a set of tests that assess the performance of the presented protocols, and finally, Section 5 draws some conclusions and outlines future work.

## 2. Industrial Communication Protocols

This section explains the protocols and connection strategies used in industrial communication, including examples of their application in the manufacturing industry. The most common and widely employed protocols in industry have been chosen [20], along with a final comparison.

### 2.1. Modbus

Modbus is an ICP used for data transfer between devices in automation and process control systems [21]. It follows a client/server model, enabling communication between devices connected across various types of buses or networks.

In order to establish a Modbus connection, one device must be configured as a server and the other as a client. By initiating a TCP/IP connection, the client sends a request to the server, which then responds by transmitting the requested information through a channel. This protocol employs various types of data messages to read or write information in compatible devices, with Modbus RTU being one of the most common methods of serial communication.

Modbus stands out for its capability to connect with a wide range of devices, and high reliability in data transmission. Furthermore, being an open-source protocol, there are a number of high-level libraries for different programming languages.

Modbus TCP, being an older industrial protocol, has limited built-in security features. It lacks encryption and authentication mechanisms by default, leaving it susceptible to unauthorized access and data manipulation. To enhance security, implementing additional measures such as Virtual Private Networks (VPNs) and application-layer encryption methods including Transport Layer Security (TLS) or Secure Sockets Layer (SSL) can effectively protect Modbus TCP communications from potential threats.

In recent years, efforts to enhance the security of Modbus TCP have led to the development of the Modbus Security protocol in 2018. Recent research focuses on strengthening Modbus TCP’s security against unauthorized access. For instance, Martins and Vidal [22] proposed authentication and authorization functions by implementing username and password based access control methods for human users with knowledge of industrial automation control systems (IACSs). Another study by de Brito and de Souza [23] introduces a testbed using the Modbus protocol to analyze cybersecurity in nuclear power plants. These advancements aim to address vulnerabilities and explore methods for securing Modbus TCP communications.

Modbus is commonly used in the manufacturing industry to transfer data between devices such as PLC and HMI (Human–Machine Interface) through the utilization of sensors. For example, Khuzyatov et al. [24] proposed a client/server approach that establishes communication between a Siemens PLC and field devices using the Modbus protocol for process control systems. In another study, Li et al. [25] developed a digital twin capable of detecting DoS attacks on Modbus TCP by using OpenPLC as a tool for virtualization of industrial control systems. In the context of IIoT communication, Folgado et al. [26] highlighted the advantages of using advanced real-time data acquisition and monitoring systems, including the practical implementation of Modbus TCP, for enhancing the performance and reliability of Polymer Electrolyte Membrane (PEM) hydrogen generators in industrial applications.

### 2.2. Profibus and Profinet

Profibus is an open field digital network standard used to interconnect process automation components such as field sensors, actuators, and PLCs in industrial environments [27]. The architecture of Profibus is built upon a client/server model.

Profibus enables the server, as the process controller, to supervise communication with clients, including drivers, motors, I/O (input/output) devices, and robots. In order to connect to a Profibus device, it is essential to have an operational and configured network, assign a device address, and establish a communication channel with a Profibus server, such as a PLC or a similar device. Once connected, the devices can exchange data and commands seamlessly.

However, with evolving industrial networking requirements, Profibus has become outdated when compared to modern fieldbus protocols such as Modbus TCP, Ethernet/IP, and Profinet. Profinet, an ethernet-based fieldbus with an open and standardized architecture, offers significant advantages over its predecessor, including faster data transmission rates, increased flexibility, and improved scalability [28]. The migration from Profibus to Profinet has enabled industrial systems to embrace the benefits of ethernet-based communication, resulting in higher efficiency, enhanced interoperability, and streamlined system integration. Profinet has now established itself as the standard for reliable and efficient communication in industrial environments.

Profibus, as an older protocol, lacks native encryption and authentication mechanisms by default, leaving it vulnerable to security breaches. In contrast, Profinet offers advanced security features, including authentication through X.509 certificates and username/password, as well as encryption using TLS or Secure Real-Time Transport Protocol (SRTP). With these measures in place, Profinet ensures data confidentiality and integrity, providing robust protection against unauthorized access and data tampering.

Profibus and Profinet are widely used in the manufacturing industry. For example, Gabor et al. [29] implemented the control of a motor by establishing communication between a Siemens S7-1200 PLC and an Eaton Variable Frequency Drive (VFD), which is monitored by an HMI that uses Profibus. Another example is presented by Kjellsson et al. [30], who examined the integration of the WISA (Wireless Interface for Sensors and Actuators) concept in wired field networks for factory automation, both in Profibus and Profinet. Additionally, Xie [31] proposed an integration architecture of Profinet and OPC UA technology, focusing on the transfer of diagnostic information to manufacturing execution systems. This highlights the significance of Profinet diagnostic information in implementing predictive maintenance strategies.

### 2.3. Ethernet/IP

Ethernet/IP is an ICP based on ethernet technology that enables real-time data transfer between devices of diverse manufacturers and technologies [32]. It is based on a client/server architecture and it is extensively deployed in control applications within production plants that demand the transmission of high-speed and high-volume data.

To implement Ethernet/IP, various devices including sensors, actuators, and controllers must be connected to a common network and communicate with each other to coordinate production operations. This requires knowing the IP addresses and names of these devices and configuring them to communicate at a specific time interval. Then, a TCP/IP connection is established with the device to facilitate the exchange of I/O messages.

Ethernet/IP provides several benefits, such as high-speed data transmission, scalability for integrating a extensive range of devices, and ease of configuration and troubleshooting. Moreover, as it is a standardized protocol, it is easier to maintain.

Ethernet/IP offers more robust security features compared to Modbus TCP. It supports authentication mechanisms, including username and password based authentication, to verify the identity of devices and users. Additionally, Ethernet/IP supports IPSec (Internet Protocol Security), which provides confidentiality, integrity, and authentication for IP-based communications. By implementing IPSec, data exchanged between devices can be encrypted, protecting it from unauthorized access.

There are multiple applications of this protocol in Industry 4.0. Bello [33] highlights its importance in automotive industry communications. Additionally, Nguyen et al. [34] propose a novel real-time communication approach based on Ethernet (RTEthernet) and present an infrastructure for controller stations with the objective of optimizing data transmission capacity across multiple machines.

### 2.4. OPC-UA

OPC-UA is a cross-platform communication protocol intended for secure and reliable data exchange in the industrial automation space [35]. An OPC architecture is composed of one or more OPC servers and OPC clients.

OPC-UA allows a constant data flow between multiple devices and control applications with limited restrictions, as well as serving as a means of communication between Supervisory Control and Data Acquisition (SCADA) applications and sensors. Bidirectional connections and persistent sessions are essential for maintaining active and continuous communication between clients and servers. In terms of capture frequency, OPC-UA is typically used to monitor a reduced set of variables (from 1 to 5).

OPC-UA is considered as the de facto communication protocol for Industry 4.0 [36]. It offers benefits such as high security, real-time transmission of large data volumes, and high scalability. Its technological independence enables compatibility with devices and platforms from various manufacturers and operating systems. However, implementing OPC-UA in environments with numerous devices can be complex and costly.

OPC-UA was designed with security as a fundamental aspect, providing comprehensive security features. It supports transport layer security protocols such as TLS/SSL, enabling encryption and authentication for secure data transmission. OPC-UA also incorporates access control mechanisms, allowing administrators to define detailed access policies for users and devices.

The adoption of this protocol offers advantages in implementing predictive maintenance in industrial machinery. For example, Tapia et al. [37] implemented a monitoring platform that used OPC-UA to acquire data from a 5-axis machining center and detect outliers in real time. Liu et al. [38] proposed a platform that integrates OPC-UA with MTConnect (manufacturing industry standard) to enhance effective communication between cyber–physical machine tools and support informed decision-making. In a study focused on IIoT, Gutierrez-Guerrero and Holgado-Terriza [39] proposed a novel mechanism for auto-configuring OPC UA systems in industrial environments. This mechanism allows for self-managed configuration over the Modbus protocol, automating the setup of the OPC-UA server from PLC devices connected to a basic Ethernet network.

### 2.5. Protocol Comparison

Table 1 provides a comparison of the five ICPs discussed above, in terms of transmission range, frequency, data rate, and security. As observed in the table, Modbus, Profibus, and Profinet protocols are well-suited for local networks, whereas Ethernet/IP and OPC-UA are better suited for wide area networks, with OPC-UA offering an unlimited transmission range. Ethernet/IP and Profinet have the highest frequency, varying between 1 and 100 MHz, and the highest data rate, capable of reaching speeds of up to 1 Gbps.

When selecting an industrial protocol for a specific application, it is important to consider factors such as network size, complexity, number of devices, and data transmission volume. These characteristics are key in determining the most appropriate protocol for the specific scenario.

Figure 2 illustrates the usage percentages of industrial protocols based on a study conducted by HMS Networks in 2019 [40]. In that particular year, industrial ethernet comprised 59% of newly installed nodes, while fieldbuses constituted 35% of industrial utilization. Ethernet/IP emerged as the predominant network, representing 15% of the total installations, whereas Modbus TCP was utilized in 4% of cases. Among fieldbuses, Profibus DP held the highest usage share, accounting for 10% of the total. Additionally, there was a noticeable growth of 30% in the adoption of wireless technologies. This increase can be attributed to the rapid progress of the IoT and the proliferation of mobile devices at the edge. The industry has recognized the numerous advantages of wireless communication, including enhanced mobility, flexibility, and connectivity. This growing adoption enables real-time data transmission, and remote monitoring and control, leading to improved operational efficiency and the creation of new applications.

## 3. Software for Industrial Data Acquisition

This section explores two data acquisition methods employed in big data environments, offering insights into the advantages and drawbacks associated with the mentioned software tools.

### 3.1. ETL Tools

Extract, Transform, Load (ETL) tools are software pieces that facilitate the process of extracting data from different sources, such as databases, files, or remote locations, transforming or manipulating the data according to predefined rules or requirements, and then loading it into a target destination, such as a data warehouse or a database. Some ETL tools provide a visual interface or programming environment that enables users to design, schedule, and automate the data integration workflows, ensuring efficient, accurate, and scalable data movement and transformation across various systems and formats.

An example of a widely used ETL tool is Apache NiFi [41], which enables the user to create data flows between different types of systems in a visual way. It provides a Graphical User Interface (GUI) where users can design data pipelines by connecting different *processors*, which are the basic building blocks of NiFi. Each *processor* offers different functionalities such as data ingestion, filtering, transformation, and data forwarding. NiFi offers a wide range of pre-built *processors*, and users can also create custom *processors* to address specific data processing requirements.

Apache NiFi offers several advantages for data management. It is highly scalable, allowing efficient handling of large data volumes. The data provenance enables tracking and auditing of data, enhancing data governance. Additionally, NiFi provides robust authentication and encryption mechanisms to ensure data security.

A limitation of Apache NiFi is its dependence and reliance on *processors*, which can impact data processing functionality based on their availability and compatibility. Complex use cases involving intricate transformations may also require additional configuration and development efforts.

### 3.2. Coding Environments

As an alternative to off-the-shelf tools, practitioners can create custom ways to extract data for machine tools using coding environments. A very popular one is Python, a versatile programming language and environment that provides a range of libraries and tools for extracting data from industrial communication protocols

As an example, and in the context of this work, Snap7 is a Python library that enables communication with PLCs through the Siemens S7 (Step7) protocol [42]. S7 is designed to work with different ICP, which are selected based on the specific communication hardware used. Commonly employed protocols for S7 include Profibus, Profinet, and Ethernet/IP.

Another commonly used library for data ingestion is pyModbusTCP [43], which provides a Python implementation of the Modbus TCP protocol. With its user-friendly classes and functions, it facilitates interaction with Modbus TCP devices, allowing easy reading and writing of data.

Python’s well-documented libraries for ICP provide developers efficient tools for extracting and processing data in big data environments. Furthermore, Python is highly favored due to its simplicity, ease of use, and open-source nature, offering the availability of libraries for various applications. However, a drawback of Python can be its performance; given its nature as an interpreted language, some Python environments do not attain as much performance from the computing machines as compiled languages. Nevertheless, a way to address this issue is to use environments such as Cython, which allow developers to write C extensions for Python, enhancing the performance of computationally intensive tasks.

## 4. Results

In this section, we present the results obtained from the empirical evaluation of three communication protocols. The testing methodology is described in Section 4.1 and Section 4.2 presents experiments evaluating the performance of the protocols in terms of acquisition rates and CPU and RAM usage. Section 4.3 provides an analysis of their usage complexity from the software perspective, including code samples from the Python libraries in use.

### 4.1. Experimental Setup

In this section, we present an overview of the testing methodology, including the industrial machines utilized, their characteristics, the variables obtained from each machine, and the approach taken to establish the connection with each protocol.

As shown in Figure 3, the extraction of variables from the PLC is carried out by establishing a direct connection with the machine using an ethernet cable. This requires connecting the cable to a device with a Python environment with the necessary libraries or a NiFi environment. The testing phase involved evaluating three protocols on three different machines. Specifically, Modbus TCP was used to retrieve variables from a broaching machine, Ethernet/IP was employed for a turning machine, and OPC-UA enabled variable extraction from a milling machine.

#### 4.1.1. Communication between Modbus TCP and a Broaching Machine

This section provides an overview of the data extraction process from an EKIN™ A218 broaching machine using the pyModbusTCP (version 0.1.10) Python library. The A218 is an electromechanical broaching machine designed for external surface applications [44]. Unlike other broaching machines, the cutting tool remains static on this machine. At the same time, the rotary indexing table, along with the workpiece, moves along the entire machine’s Z-axis, allowing for a higher cutting speed (Vc). One notable advantage of this machine is its ability to extract motor information and monitor the cutting process, providing valuable insights for analysis.

The whole system is controlled by the *FAGOR^®^ 8070* computer numerical control (CNC) software, which oversees the cutting process. In the same way, it enables the collection of data on the engine’s condition, ensuring that the data are collected at a frequency equal to or greater than the closed-loop control cycle.

When connecting to the CNC, it is important to ensure that it is within the same network range. Next, the IP address of the CNC and the port for Modbus TCP (502) should be specified. After identifying the manufacturer’s *input register addresses* and their corresponding data types, the range of addresses is defined, ensuring the starting and ending addresses for the registers to be read are provided.

Some of the most relevant monitoring variables in the A218 broaching machine are:Z-axis position.Broach length.Current slot angle.Cutting speed (Vc).

#### 4.1.2. Communication between Snap7 and a Turning Machine

In this section, we will provide an overview of the data extraction process from a turning machine’s PLC using the Snap7 (version 1.3) Python library. Specifically, we will focus on utilizing Ethernet/IP protocol to obtain data from the PLC.

The variables will be extracted from a Danobat™ TV-1500 machining center [45]. This center features a vertical turning lathe capable of performing precise operations including turning, grinding, and measuring. The vertical lathe stands out for its high dimensional stability and high damping coefficient, ensuring precision in machining processes. The machine has a *Siemens^®^ Sinumerik 840D SL* CNC integrated with a *Siemens^®^ Simatic S7-300* PLC that, among others, collects monitoring data in the form of a set of variables.

To establish a connection with the PLC using Ethernet/IP, it is necessary to ensure that the devices are in the same network range. Then, the IP address, rack number, and slot number of the PLC need to be defined. After establishing the connection, it is possible to read data from a specific *data block* (DB) by specifying the block number, starting address, and data size. For this purpose, it is essential to consult the manufacturer regarding the definition of the DBs.

Some of the most relevant monitoring variables in the TV-1500 are:X- and Z-axes position.Spindle axis position.Spindle rotational speed.Spindle power.

#### 4.1.3. Communication between OPC-UA and a Milling Machine

This section outlines the process of extracting data from an Ibarmia™ THR 16 [46] five-axis milling machine PLC using OPC-UA. The THR16 is a machining center that combines different technologies in a single machine: milling, drilling, turning, gear cutting, and grinding. It is considered a *Multiprocess* machine, as it allows machining different types of pieces with the same tool-set. This reduces the quantity of parts that must be manufactured in batches, which shortens the production life cycle and reduces the amount of shifts between machines in a factory.

The machine is also equipped with a *Siemens^®^ Sinumerik 840D SL* CNC integrated with a *Siemens^®^ Simatic S7-300* PLC, which collects monitoring data. In order to establish a connection between the PLC and our system, we will utilize an OPC-UA environment with one client and one server. Our computer will function as the OPC client, while the PLC of the five-axis milling machine will act as the OPC server. In terms of security, two server certificates were configured and installed on the client side to provide an additional layer of security.

Some of the most relevant monitoring variables in the THR16 are listed below. Load, power, and rotational speed are measured for the X, Y, and Z linear axes and the A and C rotary axes:X-, Y-, Z-, A-, and C-axes load.X-, Y-, Z-, A-, and C-axes power.X-, Y-, Z-, A-, and C-axes rotational speed.

### 4.2. Performance Evaluation

In this section, we present a set of tests that assess the performance of the industrial protocols, focusing on two key aspects: sampling rate and CPU and RAM usage. By assessing the sampling rate, we aim to determine the protocols’ efficiency in transmitting and receiving data. Additionally, the evaluation of CPU and RAM usage provides insights into the impact of these protocols on the computational resources of the system.

Considering the security aspects of these protocols, our tests were conducted on a virtual machine hosted by a private cloud within the CFAA premises [47]. Communication between the manufacturing machines and the data center is established through a local ethernet network. To ensure a secure connection, a VPN was employed, allowing authorized CFAA users exclusive access to the manufacturing machines.

The virtual machine used for the tests has the following specifications:Virtual Cores (CPUs): 2;CPU frequency: 2.1 GHz;RAM Memory: 8 GB;Disk space: 256 GB;Operating system: Ubuntu 20.04.

#### 4.2.1. Sampling Rates

In IIoT systems, achieving high performance connectivity is essential, as low-latency requirements play a critical role in enabling real-time decision-making in data extraction applications. To this end, in this section we focus on evaluating the sampling rate of industrial protocols.

In the first test, we measured the time taken by each protocol to retrieve batches of different sizes (i.e., a distinct set of variables from the corresponding machine was retrieved for each batch). Even if the variables mentioned in Section 4.1 are the most pertinent ones for the use case, we gathered performance metrics for larger batches to test the scalability of the platform.

The sampling rates obtained by extracting different amounts of variables using each ICP are depicted in Figure 4. In this analysis, all acquired variables were of float data type and occupied 4 bytes each. The results were derived from five separate measurements, where a computer was directly connected to the machining center, as shown in Figure 3. Among the three protocols, Modbus demonstrated the highest sampling rate, followed by OPC-UA, while Ethernet/IP exhibited the longest processing time. It is important to note that the testing did not include a larger number of variables to avoid overloading each machine’s PLC. To ensure the optimal performance of the machine’s PLC, it is advisable to reduce the acquisition frequency or the number of variables per batch. Additionally, implementing effective data compression techniques or employing a data aggregation strategy can significantly reduce the data transmitted to the PLC, thereby preventing overload.

#### 4.2.2. CPU and RAM Usage

The evaluation of CPU and RAM usage provides valuable insights into the influence of communication protocols on the computational resources of the system, particularly in terms of data acquisition delays.

To further investigate this aspect, we performed an analysis of the CPU and memory consumption of the computer being used to capture data. The test involved retrieving batches of 50 variables and was carried out using the Linux *top* command for a duration of 2 min while all resources were actively running. In order to assess the most critical scenario, we conducted the analysis using batches of 50 variables. During the testing, we observed minimal variation in CPU and RAM consumption when requesting 1 to 50 variables.

Results of this test are presented in Table 2. From the table, it can be observed that the CPU and RAM consumption levels are relatively low, indicating that the ETL or programming environments do not become a bottleneck. Among the protocols tested, S7 showed the highest CPU consumption at 45%, while OPC-UA consumed the most RAM memory at 37% due to the inclusion of NiFi *processors*. These findings suggest that the overall behavior of variable extraction is not significantly impacted, as the CPU and RAM consumption remains below 50% in all cases.

### 4.3. Development Complexity

In this section, we will provide some discussion about the complexities of data extraction from the software perspective, including some details with code snippets about the usage of Python libraries. We will also discuss the custom processors that have been employed to establish communication with OPC-UA.

With regard to Modbus, we present in Listing 1 a snippet summarizing the usage of pyModbusTCP to extract data. In the first line of the code, a Modbus client is created using the specified PLC IP address and port (502) to establish the connection. The second line of the code retrieves eight input registers starting from register address 1. Next, the third line combines the values of the last two 2 registers, where the data of our value are stored, into a 32-bit value using the “<<” operator.

**Listing 1.** Code snippet using pyModbusTCP to extract a variable from a PLC.
c = ModbusClient(host=PLC_IP, port=502, unit_id=1, auto_open=True)

regs = c.read_input_registers(1,8)

last_2bytes_registers = (regs[6] << 16) + regs[7]

4bytesregister = struct.pack(’>L’, last_2bytes_registers)

value_float = struct.unpack(’>f’, 4bytesregister)[0]


The fourth line packs the 32-bit value as a 4-byte binary data string using the *struct.pack* function. This function converts the value into a binary format using the “>L” format specifier. Finally, the binary data, which are stored as a 32-bit floating-point value, are unpacked and assigned to a float variable using the *struct.unpack* function with the “>f” format specifier.

Regarding the S7 protocol, it retrieves variables via Ethernet IP at a slower pace compared to Modbus and OPC-UA. However, the code required for communication is much simpler.

To obtain the values of variables stored in the PLC using Ethernet/IP, it is necessary to know their addresses and structure. These variables are stored in DBs. For example, a variable defined in the PLC may be referred to as DB24.DBW32. Here, DB24 indicates that the variable is located in data block 24, while DBW32 indicates that it starts at an address of 32 bytes and has a length of 2 bytes (16 bits), denoted by the letter “W” (Word). Boolean variables are represented by an “X” and occupy 1 bit, while “B” (Byte) variables have a length of 1 byte (8 bits) and “D” (Double Word) variables have a length of 4 bytes (32 bits).

As demonstrated in Listing 2, we are reading the address DB129.DBD0 from the PLC, which corresponds to the X-axis position of the turning machine. Once the connection with the PLC is established, a data point can be read from the specific data block by specifying its number, starting address, and data size. In this case, the variable is a double word (D) hosted in DB129, starting at byte 0. The data point is returned as a byte array and then converted to a decimal number for further processing.

**Listing 2.** Code snippet using Snap7 to extract a variable from a PLC.
db_number = 129

start_byte = 0

client = snap7.client.Client()

client.connect(PLC_IP, rack=0, slot=2)

byte_array = client.db_read(db_number, start_byte, 4)

value = snap7.util.get_real(byte_array,0)


Finally, and with regard to OPC-UA, we utilized custom NiFi *processors* developed by Zylk [48], a third party, to set communication with the THR16. The monitored variables are extracted and processed by NiFi (version 1.15.3), which creates data streams through *processors*. As shown in Figure 5, two custom *processors* have been configured to facilitate the data flow between the OPC Server and NiFi. The *ListOPCNodes processor* is responsible for listing the nodes we have specified, while the *GetOPCData processor* retrieves the data from those nodes.

The development complexity of data extraction varies among the discussed protocols. Modbus and S7 protocols require basic programming skills and knowledge of the protocols, particularly in Python, to write code for data extraction. In contrast, utilizing OPC-UA requires familiarity with NiFi and proficiency in using its interface to connect processors effectively. Once configured, the graphical interface simplifies the configuration process, providing a user-friendly option for visual setup.

To summarize, Table 3 provides an overview of the features offered by the protocols used in our experiments, including frequency, code complexity, and required infrastructure. This information aims to assist other experts in selecting a suitable communication protocol for their specific IIoT application.

## 5. Conclusions

This work has presented a study of the performance for three industrial communication protocols: Modbus, Snap7, and OPC-UA. A description of the protocols has been presented, along with some software tools able to use them. An empirical evaluation has been conducted with three machine tools in order to assess the performance of the protocols in a real environment. Results have shown that Modbus provides the best latency figures, and that communication has different complexities depending on the used protocol, from the software perspective.

The research conducted enabled the answering of the proposed research questions. The differences in terms of performance and usability complexity from a software perspective are summarized in Table 3. Similarly, various performance tests were conducted during the research, allowing for the evaluation and comparison of the studied ICPs.

Future work will span in two directions. Firstly, we will examine other features of industrial communication protocols, such as reliability and throughput. Secondly, we will extend this study to other protocols, including those more oriented to IIoT environments, such as Advanced Message Queuing Protocol (AMQP) and Message Queue Telemetry Transport (MQTT).

## Figures and Tables

**Figure 1 sensors-23-05694-f001:**
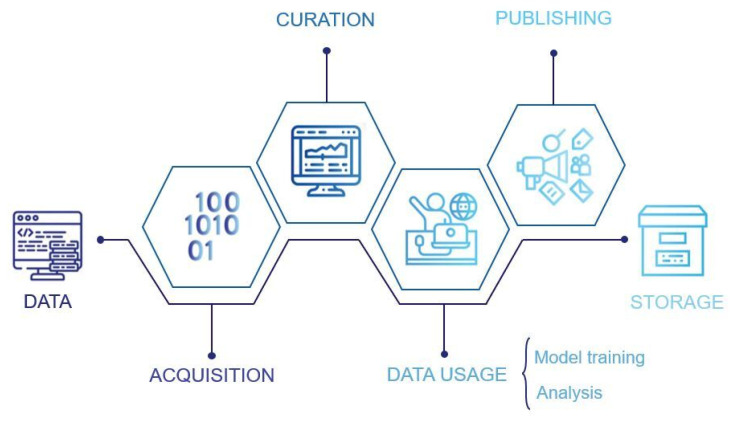
Representation of the typical stages in a data analysis platform.

**Figure 2 sensors-23-05694-f002:**
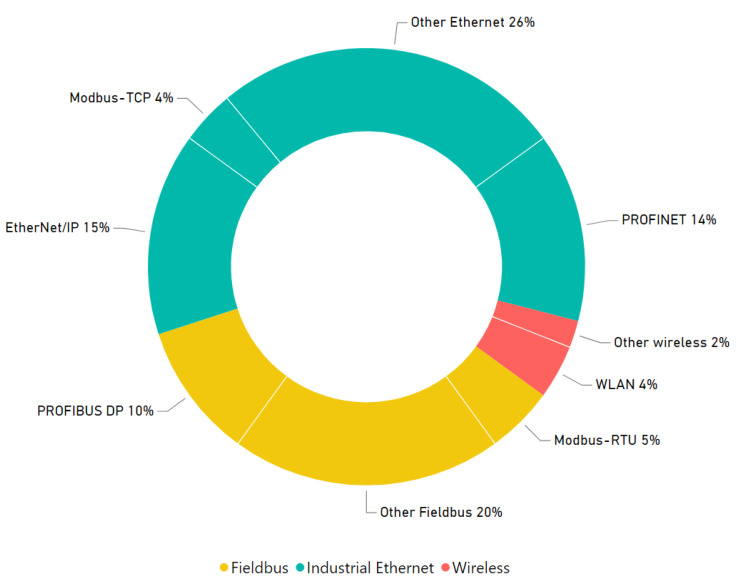
Use of industrial communication protocols in 2019 [40].

**Figure 3 sensors-23-05694-f003:**
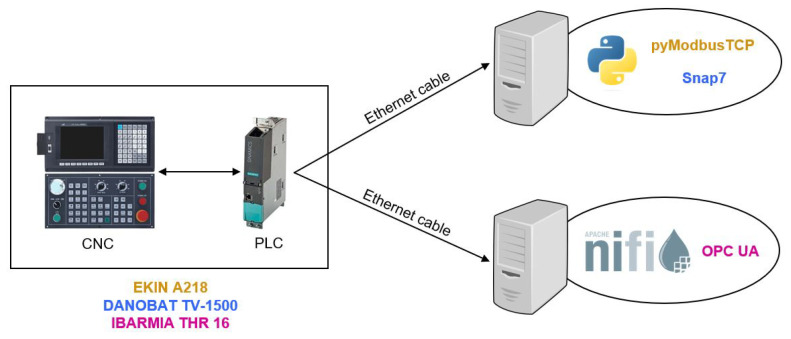
Connection between manufacturing machines and communication protocols.

**Figure 4 sensors-23-05694-f004:**
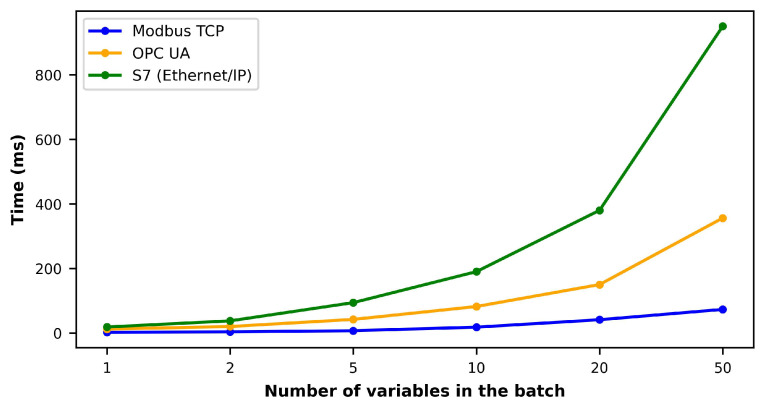
Comparison of sampling rates for each protocol with different amounts of variables extracted from the PLC.

**Figure 5 sensors-23-05694-f005:**
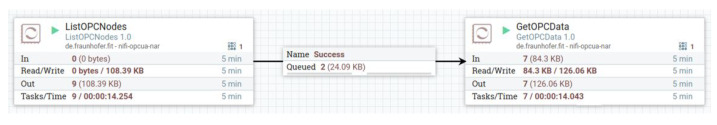
NiFi custom *processors* used to retrieve data from OPC-UA.

**Table 1 sensors-23-05694-t001:** Industrial communication protocols comparison.

Protocol	Range	Frequency	Data Rate (Mbps)	Security
Modbus	Local/Wide Area Networks	1–1000 Hz	Up to 0.1 Mbps	Low
Profibus	Local/Wide Area Networks	1–16 MHz	Up to 12 Mbps	Low
Profinet	Local/Wide Area Networks	1–100 MHz	Up to 1000 Mbps	High
Ethernet/IP	Wide Area Networks	1–100 MHz	Up to 1000 Mbps	Medium/High
OPC-UA	Unlimited	1–10 kHz	Up to 100 Mbps	High

**Table 2 sensors-23-05694-t002:** CPU and memory usage for batches of 50 variables.

Protocol	CPU Usage (%)	RAM Usage (%)
Modbus TCP	27%	24%
S7 (Ethernet/IP)	45%	26%
OPC-UA	18.4%	37%

**Table 3 sensors-23-05694-t003:** Summary of evaluation of the protocols.

Protocol	Frequency	Code Complexity	Infrastructure
Modbus	500–700 Hz	Simple	Python environment with pyModbusTCP library
Ethernet/IP	50–60 Hz	Simple	Python environment with Snap7 library
OPC-UA	80–150 Hz	Medium	Nifi environment with custom OPC-UA processors

## Data Availability

No new data were created.

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
