# Peer review of "Assessing Industrial Communication Protocols to Bridge the Gap between Machine Tools and Software Monitoring"

_sensors, 2023, doi:10.3390/s23125694_

Round 1

Reviewer 1 Report

The presented review study is topical for IIoT. Nevertheless, the specified revision is needed to properly shape and relate the study and its key findings.

- The study considers particular communication protocols. Their names should be listed in the keywords section.

- Introduction includes much content that is close to “Related Work”. Please, revise Introduction (practical needs, research problem definition, contribution, methodology, etc.) and add a separate section “Related Work”.

- The evaluation of industrial communication protocols does not consider such important requirements on communication efficiency as data security and smart communication, e.g., see https://doi.org/10.22364/bjmc.2016.4.4.28

- Both security and smartness requirements need attention to edge computing, which is an important paradigm of IoT (likely, fog computing can be considered as well), e.g., see https://www.fruct.org/publications/volume-29/acm29/files/Bes.pdf

Basically, the raw sensor data are not sent but some data processing is implemented locally (or on intermediatory devices) before the data transmission, so increasing security and communication efficiency.

- Indeed, industrial requirements to efficiency lead to the use of wired communication protocols. Nevertheless, the IoT progress with mobile devices at the edge leads to more active use of wireless communication. Some discussion on this issue must be presented.

- Key experimental findings with particular estimates need summarization in a table. This summary result should support other experts to select a communication protocol for the use in a given IIoT application.

Author Response

We sent to the editor a ZIP file containing the final version of the manuscript, where all the modifications have been incorporated into the text. Additionally, within the ZIP file, you will find a PDF version of the manuscript with the changes clearly highlighted in BLUE for easy reference. Please see the attachment.

Reviewer 2 Report

The authors assessed the performance and complexity of OPC-UA, Modbus, and Ethernet/IP in the industrial environment. The evaluation focuses on their effectiveness in the real-time acquisition of machine monitoring data and compares their performance from the software perspective. 

The paper emphasizes the importance of empirically evaluating the performance of industrial communication protocols. It highlights the process of extracting monitoring variables from the machine’s PLC and assesses their performance in terms of latency, CPU usage and RAM usage.

However, the authors lack to provide the necessary considerations for the evaluation process and a detailed explanation regarding how they derived results from the measurements. 

o  This paper lacks the necessary consideration to evaluate the performance of the protocols.

o  It lacks a detailed explanation of how the authors derived the results from the measurements.

o  The test was conducted using 50 batches of variables to evaluate the performance. However, the authors could have varied the number of batches of variables to enhance the analysis.

o  The authors claim that large numbers of variables are avoided to prevent overloading the machine's PLC. Are there any alternative methods to prevent overloading the machine's PLC?

o  The introduction can be improved by adding a few more recent papers on the performance evaluation of industrial communication protocols. Also, some references need to be explained with more relevant details.

o  The research design can be improved, the authors could assess the performance based on different computer configurations.

o  The authors’ representation of CPU and RAM usage for different protocols in Table 2 lacks a clear articulation of the impact of these resource allocations on the overall results. 

o  In general, the quality of the paper can be improved, an in-depth technical explanation is required to further improve the paper.

Moderate editing of English language required.

Author Response

(The authors gave the same response as above.)

Reviewer 3 Report

The topic of the manuscript is interesting given the relevance that communication protocols and software have in different scientific and industrial fields; in addition, this topic fits the scope of the Journal. After a careful revision, the following comments are provided for the enhancement of the manuscript.

“Monitoring” could be added as keyword, if the authors agree.

PROFIBUS is a fieldbus very used in industry; however, it is a bit old in comparison with Modbus TCP, Ethernet/IP and OPC-UA. In fact, it is being replaced by Ethernet-based fieldbuses, like the aforementioned and PROFINET. The latter one is slightly commented in line 123. Therefore, this reviewer suggests enlarging the information about PROFINET in the subsection 2.2 and changing its title to PROFIBUS and PROFINET for a more adequate contextualization of the considered fieldbuses.

Some of the tools used in the approach are of open-source nature, such as the Python libraries. This is a positive feature that should be emphasized as strength of the paper in the subsection 3.2.

Regarding Modbus TCP (subsection 2.1) some aspects are commented to enhance the provided description. To begin with, recent publications which deal with the studied protocols in the modern context of IIoT could be cited to highlight the relevance of such protocols, for example, the following ones:

-        Data acquisition and monitoring system framed in Industrial Internet of Things for PEM hydrogen generators. Internet of Things 2023, 22, 100795. https://doi.org/10.1016/j.iot.2023.100795

-        Automatic Configuration of OPC UA for Industrial Internet of Things Environments. Electronics 2019, 8, 600. https://doi.org/10.3390/electronics8060600

Additionally, there is a release of this protocol which is supposed to have enhanced security features, namely, the so-called Modbus Security protocol, published in 2018. Moreover, developments about security aspects of Modbus constitute a relevant research scope as witnessed by recent publications such as the following ones:

-        Enhanced Modbus/TCP Security Protocol: Authentication and Authorization Functions Supported. Sensors 2022, 22, 8024. https://doi.org/10.3390/s22208024

-        Development of an Open-Source Testbed Based on the Modbus Protocol for Cybersecurity Analysis of Nuclear Power Plants. Appl. Sci. 2022, 12, 7942. https://doi.org/10.3390/app12157942

When describing the protocol OPC-UA, the authors could mention that it is considered as the de facto communication protocol for Industry 4.0.

In the fourth section, the specific model of the PLC is not found. From figure 3, one can suppose that is manufactured by Siemens but a clear indication of this details is required.

The abbreviation DB stands for data block, which is the structure for data storage within the PLC memory. This abbreviation is defined twice, in line 287 and in line 371, being required only the first one.

The title of table 1 must be placed before the table in itself.

Some discussion about the difficulty degree of programming the data access in the three different protocols would enrich the last section. For example, for Modbus and for Snap7, the programming is via code, whereas for OPC-UA it is using a graphical interface. For the interested reader, the point of view of the authors could be constructive.

Author Response

We sent to the editor a ZIP file containing the final version of the manuscript, where all the modifications have been incorporated into the text. Additionally, within the ZIP file, you will find a PDF version of the manuscript with the changes clearly highlighted in BLUE for easy reference.

Round 2

Reviewer 3 Report

The new version of the paper has been improved by taking into consideration the suggestions of reviewers. Congratulations to the authors for their efforts.